# Immunohistochemical Glucagon-like Peptide-1 Receptor Expression in Human Insulinomas

**DOI:** 10.3390/ijms242015164

**Published:** 2023-10-13

**Authors:** Tiina Vesterinen, Elina Peltola, Helena Leijon, Päivi Hannula, Heini Huhtala, Markus J. Mäkinen, Lasse Nieminen, Elina Pirinen, Mikko Rönty, Mirva Söderström, Pia Jaatinen, Johanna Arola

**Affiliations:** 1Department of Pathology, HUSLAB, HUS Diagnostic Center, Helsinki University Hospital, University of Helsinki, 00290 Helsinki, Finland; tiina.vesterinen@helsinki.fi (T.V.); helena.leijon@hus.fi (H.L.); mikko.ronty@hus.fi (M.R.); johanna.t.arola@helsinki.fi (J.A.); 2Institute for Molecular Medicine Finland (FIMM), Helsinki Institute of Life Science (HiLIFE), University of Helsinki, 00290 Helsinki, Finland; 3Faculty of Medicine and Health Technology, Tampere University, 33014 Tampere, Finland; elina.peltola@tuni.fi (E.P.); paivi.hannula@pirha.fi (P.H.); 4Department of Internal Medicine, Tampere University Hospital, 33520 Tampere, Finland; 5Endocrinology, Department of Internal Medicine, Tampere University Hospital, 33520 Tampere, Finland; 6Faculty of Social Sciences, Tampere University, 33014 Tampere, Finland; heini.huhtala@tuni.fi; 7Department of Pathology, Research Unit of Translational Medicine, Oulu University Hospital, University of Oulu, 90220 Oulu, Finland; markus.makinen@oulu.fi; 8Fimlab Laboratories, Pathology Department, Tampere University Hospital, 33520 Tampere, Finland; lasse.nieminen@fimlab.fi; 9Department of Clinical Pathology, Kuopio University Hospital, 70029 Kuopio, Finland; elina.pirinen@pshyvinvointialue.fi; 10Department of Pathology, Turku University Hospital, 20521 Turku, Finland; mirsod@utu.fi; 11Division of Internal Medicine, Seinäjoki Central Hospital, 60220 Seinäjoki, Finland

**Keywords:** glucagon-like peptide-1 receptors, insulinoma, neuroendocrine tumour, immunohistochemistry

## Abstract

Insulinomas are rare functional pancreatic neuroendocrine tumours, which metastasize in 10% of cases. As predicting the prognosis can be challenging, there is a need for the determination of clinicopathological factors associated with metastatic potential. The aim of this study is to evaluate the glucagon-like peptide-1 receptor (GLP-1R) expression in insulinomas and to analyse its association with clinicopathological features and patient outcome. This retrospective study involves pancreatic tumour tissue samples from fifty-two insulinoma patients. After histological re-evaluation, formalin-fixed paraffin-embedded tissue samples were processed into tissue microarrays and stained immunohistochemically with a monoclonal GLP-1R antibody. Forty-eight of the forty-nine (98%) non-metastatic tumours expressed GLP-1R, while one non-metastatic, multiple endocrine neoplasia type 1 (MEN1)-related tumour and all three of the metastatic tumours lacked GLP-1R expression. The lack of GLP-1R expression was associated with impaired overall survival, larger tumour diameter, higher Ki-67 PI and weaker insulin staining. Somatostatin receptor 1–5 expression did not differ between GLP-1R-positive and GLP-1R-negative insulinomas. In conclusion, the lack of GLP-1R expression is associated with metastatic disease and impaired survival in insulinoma patients. Thus, GLP-1R expression could be a useful biomarker in estimating the metastatic potential of the tumour and the prognosis of surgically treated patients.

## 1. Introduction

Insulinomas are rare functional pancreatic neuroendocrine tumours (PanNETs). The excessive insulin secretion by the tumour leads to repeated episodes of hypoglycaemia [1,2]. Most insulinomas are non-metastatic and can be cured by surgery [3,4]. Only 10% of insulinomas metastasize, but they are associated with a significantly impaired survival since the clinical management of a metastatic disease is challenging [1,5]. The occurrence of insulinoma tumour recurrence or metastasis cannot be reliably predicted with the current biomarkers. Thus, molecular parameters that distinguish non-metastatic insulinomas from those with a metastatic potential are needed.

Previous research has indicated multiple clinical, histopathological and genetic differences between metastatic and non-metastatic insulinomas, which could be of help in guiding the therapy and predicting the prognosis of patients [6]. One of the suggested markers is glucagon-like peptide-1 receptor (GLP-1R), a member of the glucagon receptor family with a broad expression in pancreatic β-cells and a less abundant expression in the acinar cells [7]. It has previously been shown that GLP-1R expression is more frequent in insulinomas than in other neuroendocrine tumours [8,9]. Moreover, non-metastatic insulinomas tend to express high densities of GLP-1Rs, whereas their expression is rare in metastatic insulinomas [10]. Thus, GLP-1R is a promising biomarker for differentiating between non-metastatic insulinomas and tumours with a more aggressive behaviour.

The objective of this study is to analyse the immunohistochemical GLP-1R expression and its association with clinicopathological features and patient outcome in a national series of insulinomas [11,12].

## 2. Results

### 2.1. Patient Characteristics

We have previously studied the immunohistochemical somatostatin receptor expression in forty-nine non-metastatic and three metastatic insulinomas diagnosed in Finland during 1980–2010 [11]. The formalin-fixed paraffin-embedded primary tumour tissue samples from this cohort were obtained from the five Finnish University Hospitals through local biobanks (Helsinki Biobank, Finnish Clinical Biobank Tampere, Auria Biobank, Biobank Borealis of Northern Finland, and the Biobank of Eastern Finland). The characteristics of this cohort (*n* = 52) and the whole Finnish insulinoma cohort (*n* = 79) are described in detail in our previous publications [11,12]. Table 1 summarises the patient data.

### 2.2. GLP-1 Receptor Expression

All but four tumours showed at least weak expression of GLP-1R (Figure 1). In twenty-five (48%) tumours, 100% of tumour cells were intensively stained (staining intensity 3+). Of the four insulinomas that lacked the expression of GLP-1R, three were metastatic and one was non-metastatic, but the patient had MEN1 syndrome.

### 2.3. GLP-1 Receptors, Tumour Size and Ki-67 Proliferation Index (PI)

The lack of GLP-1R expression was associated with a larger tumour diameter and a higher Ki-67 PI. The median diameter of GLP-1R-negative tumours was significantly larger, 38 (range 15–60) mm, compared to 14 (range 5–40) mm in GLP-1R-positive tumours, *p* = 0.004. The median Ki-67 PI of GLP-1R-negative vs. positive tumours was 1.8% (range 0.4–16.1) vs. 0.4% (range 0.1–4.6), respectively, *p* = 0.022.

### 2.4. GLP-1 Receptors and Insulin Expression

The lack of GLP-1 receptor expression was associated with a weaker immunohistochemical staining for insulin. All three of the insulinomas, in which the insulin staining was considered weak, also lacked the expression of GLP-1R. The difference in insulin staining between the tumours expressing GLP-1R and those lacking GLP-1R expression was statistically significant (*p* < 0.002).

### 2.5. GLP-1R versus SSTR Expression

There was no statistically significant difference in the SSTR1–5 expression between GLP-1R-positive and GLP-1R-negative insulinomas. All GLP-1R-negative and most (69%) GLP-1R-positive tumours expressed SSTR2 subtype (*p* = 0.244). SSTR1 was expressed in three (75%), SSTR3 in two (50%), SSTR4 in 0% and SSTR5 in 0% of the GLP-1R-negative tumours, compared to eleven (23%) (*p* = 0.055), fifteen (31%) (*p* = 0.589), 0% and three (6%) (*p* = 1.000) in GLP-1R-positive tumours, respectively.

### 2.6. GLP-1 Receptors and Metastatic or MEN1-Related Insulinoma

The lack of GLP-1R expression was associated with a metastatic disease (*p* < 0.001). All sporadic, non-metastatic insulinomas expressed GLP-1R, whereas all three of the metastatic insulinomas lacked the expression of GLP-1R. Our cohort included two MEN1-related insulinomas: one of them showed strong expression of GLP-1R, and the other one lacked the expression of GLP-1R, as described above.

### 2.7. GLP-1 Receptors and Patient Outcome

The lack of GLP-1R expression was associated with a significantly impaired overall survival in patients with sporadic insulinomas (Figure 2). Two of the three patients with metastatic, GLP-1R-negative insulinomas died 1 and 4 years after the primary surgery, while the third patient was alive at the end of the follow-up (over 6 years after the surgery). The GLP-1R-negative, MEN1-related insulinoma did not metastasize during the follow-up of over 20 years.

None of the GLP-1R-positive insulinomas metastasized during the median follow-up of 11 years (range 0.2–32) after the primary surgery. In one patient, however, a local tumour recurrence was detected and reoperated on 10 years after the primary enucleation of a single tumour, as described earlier [12].

## 3. Discussion

In this study we analysed the immunohistochemical GLP-1R expression of fifty-two insulinomas in association with clinicopathological variables. Our main findings were that non-metastatic insulinomas express GLP-1R and the lack of GLP-1R expression is associated with metastatic disease and impaired overall survival in patients with sporadic insulinomas.

Insulinomas are known to express GLP-1 receptors. Reubi and Waser utilised in vitro receptor autoradiography that identifies and quantifies peptide receptor proteins, and observed that over 90% of insulinomas (*n* = 27) showed an extremely high GLP-1R density [8]. Later, Waser et al. reported that in their series of thirty-eight insulinomas, all tumours classified as ‘benign’ (*n* = 31) expressed GLP-1R with immunohistochemistry, while only one insulinoma classified as ‘malignant’ (*n* = 7) showed positivity [10]. Our study confirms these findings.

On the other hand, the association between GLP-1R expression and patient outcome is less studied. Cases et al. used immunohistochemistry and reported that in their series of sixteen insulinomas, 81% showed GLP-1R expression, but the expression was not significantly associated with survival [9]. However, their study also included other PanNETs, like gastrinomas and nonfunctioning tumours, and the association was studied in the whole tumour series.

Ki-67 PI is a cornerstone in the histopathological classification of PanNETs since it is used to grade the tumour as G1, G2 or G3, reflecting the prognosis of the disease [4]. We showed that the lack of GLP-1R expression is associated with a higher Ki-67 PI. Moreover, we showed an association between the lack of GLP-1R expression and a larger tumour diameter and weaker insulin staining. This is in line with previous findings, as a higher Ki-67 PI, a larger tumour diameter and a scarce insulin expression have all been associated with aggressive insulinomas [6]. To our knowledge, none of the previous studies have evaluated the immunohistochemical GLP-1R expression in relation to Ki-67 PI, tumour size or insulin expression.

In our series, the SSTR1–5 expression did not differ between GLP-1R-positive and GLP-1R-negative insulinomas. As shown earlier by Reubi and Waser, most GLP-1R-positive tumours showed SSTR2, while the expression of SSTR1, 3, 4 and 5 was lower [8].

This cohort included two MEN1-related insulinomas, of which one expressed GLP-1R and the other one did not. To our knowledge, the immunohistochemical GLP-1R expression in MEN1-related insulinomas has not been studied before. Antwi et al. observed high accuracy of ^68^Ga-exendin-4 PET/CT imaging in the localisation of MEN1-related insulinomas, which could indicate the presence of GLP-1R overexpression in most MEN1-related insulinomas [13]. The study by Antwi et al., however, included only six consecutive insulinoma patients with a genetically proven MEN1 mutation.

The overexpression of GLP-1 receptors in most insulinomas is increasingly utilised in PET/CT imaging using radiolabelled GLP-1R analogues. GLP-1R PET/CT imaging has proved to be a highly sensitive method for localising small, non-metastatic insulinomas [14]. In addition to serving as a target for molecular imaging, the significant difference in the GLP-1R expression between non-metastatic and metastatic insulinomas suggests that the immunohistochemical GLP-1R expression may prove to be, upon follow-up validation, a beneficial prognostic factor in estimating the metastatic potential of the tumours and thus the prognosis of surgically treated patients with insulinoma.

The overexpression of GLP-1Rs in insulinomas also shows high potential for tumour targeted therapy with radiolabelled GLP-1R-selective analogues. Recent findings suggest that GLP-1R-targeted radionuclide therapy with ^111^In-labeled exendin-4 derivatives could be useful in the future treatment of selected patients with an insulinoma [15,16]. Although metastatic insulinomas often seem to lack GLP-1R expression, and non-metastatic insulinomas can usually be removed surgically, the GLP-1R-targeted radionuclide therapy could become a novel treatment strategy for patients with a GLP-1R-positive insulinoma, when curative surgery in not an option.

The major limitation of this study was the small number of patients with metastatic or MEN1-related insulinomas. As only two patients had a MEN1-related insulinoma, one expressing and the other one not expressing GLP-1R, we were not able to draw conclusions from the GLP-1R expression of MEN1-related insulinomas in general. Another limitation is the lack of GLP-1R PET/CT imaging results in this cohort. As GLP-1R imaging was not available in Finland during the study period (1980–2010), we could not analyse the association of immunohistochemical GLP-1R expression and the results of GLP-1R PET/CT imaging.

The main strength of this study was the relatively large sample of fifty-two insulinomas with comprehensive clinical data and long-term follow-up of the patients. Despite the small number of metastatic insulinomas, our study clearly indicated a significant difference in the GLP-1R expression between metastatic and non-metastatic insulinomas. Moreover, all tumours were re-evaluated by a pathologist with special expertise in endocrine pathology. Our TMAs included punches both from the middle of the tumour as well as from the tumour border.

In conclusion, all sporadic, non-metastatic insulinomas in our study expressed GLP-1R, while the lack of GLP-1R was associated with a metastatic disease and impaired survival. Furthermore, the conclusive confirmation that a lack of GLP-1R expression could be used as a negative prognostic marker in insulinomas would require additional studies investigating a higher number of metastatic insulinomas.

## 4. Materials and Methods

### 4.1. Tissue Microarray Construction

The TMA construction has been described previously [11]. Briefly, two representative 1 mm cores from the middle of the tumour and from the tumour border were punched whenever possible, considering the tumour size. The TMAs were constructed in the biobanks using a TMA Grand Master (3D HISTECH, Budapest, Hungary) or a Galileo TMA CK4500 (Isenet, Milan, Italy) microarrayer.

### 4.2. Immunohistochemistry

Immunohistochemical stainings were performed with a semiautomated AutoStainer instrument (Lab Vision Corp., Fremont, CA, USA). After deparaffinisation, a heat-induced antigen retrieval in pH 9 was used before incubating the TMA sections with a primary GLP-1R antibody (diluted 1:25, clone 3F52, Developmental Studies Hybridoma Bank/Novo Nordisk A/S, University of Iowa, Iowa City, IA, USA). EnVision FLEX+ Mouse (LINKER) (Dako, Agilent Pathology Solutions, Santa Clara, CA, USA) was used for signal amplification. Antibody binding was visualised with EnVision FLEX kit (Dako). The methods for staining and scoring of Ki-67, somatostatin receptors (SSTR) 1–5 and insulin have been described previously [11].

### 4.3. Scoring of the Staining Results

Stained GLP-1R slides were digitised with a Pannoramic 250 Flash III slide scanner (3DHISTECH, Budapest, Hungary) using a calibrated linear colour model, brightfield scanning mode, and a 20× objective. Using the CaseViewer software (version 2.4.0.119028; 3DHISTECH, Budapest, Hungary), J.A. and T.V. performed the scoring manually. The TMA spot with the strongest staining pattern per tumour was chosen for evaluation and only membranous staining was considered. The proportion of stained tumour cells was expressed in percentages with increments of 10 (i.e., 0%, 10%, 20%, … 100% of tumour cells stained). The intensity of staining was analysed using the following scoring system presented by Körner et al. [17]: 1+ = faint staining at 100× magnification; 2+ = strong staining at 100× magnification, not entire circumference of tumour cell membranes stained at 400× magnification; 3+ = strong staining at 100× magnification, entire circumference of tumour cell membranes stained at 400× magnification (Figure 3). The GLP-1R expression was considered strong (score 3) if the staining intensity was 3+ and ≥50% of the tumour cells were stained. The intermediate score (2) was given if the staining intensity was 3+ but <50% of the tumour cells were stained, or if the staining intensity was 2+. The weak score (1) was given if the staining intensity was 1+, and negative (0) if no membranous staining was observed.

### 4.4. Statistical Analysis

The statistical analyses were conducted with the IBM SPSS Statistics for Windows, Version 28.0 (IBM Corp., Armonk, NY, USA). The data are presented as median (minimum–maximum) for continuous variables, and a number (%) for categorical variables. The median tumour diameter and Ki-67 index were compared between the GLP-1 receptor-positive and negative tumours with the Mann–Whitney U test. The association of GLP-1R expression with the expression of insulin and SSTR1-5, and with a metastatic disease was analysed with the Fisher Exact test. The overall survival of patients with GLP-1R-positive vs. GLP-1R-negative tumours was compared using Kaplan–Meier analysis with the log-rank test. In all analyses, a two-sided *p* value below 0.05 was considered statistically significant.

### 4.5. Ethical Considerations

This study was conducted in accordance with the Declaration of Helsinki. Informed consent was waived, because the Finnish Biobank Act provides a lawful basis for research use of biobanked samples. The Regional Ethics Committee of the Tampere University Hospital catchment area (protocol code R15175), the Scientific Steering Committees of the Finnish biobanks (project numbers BB_2020-0077, 792/2020 and BB_2020_4007), the Finnish Institute for Health and Welfare (THL/1515/5.05.00/2019), and the Finnish Medicines Agency (FIMEA/2023/001393) reviewed and approved the study protocol. The University Hospitals of Helsinki, Kuopio, Oulu, Tampere and Turku, and the Finnish Population Register Centre yielded permissions for the use of data from their registers.

## Figures and Tables

**Figure 1 ijms-24-15164-f001:**
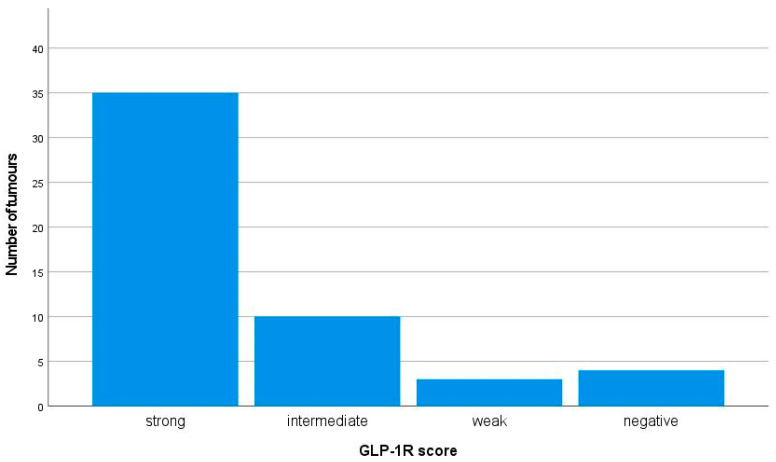
Immunohistochemical membranous GLP-1R expression in 52 insulinomas. The GLP-1R expression was considered strong if the staining intensity was 3+ and ≥50% of the tumour cells were stained. The intermediate score was given if the staining intensity was 3+ but <50% of the tumour cells were stained or if the staining intensity was 2+. The weak score was given if the staining intensity was 1+, and negative if no membranous staining was observed.

**Figure 2 ijms-24-15164-f002:**
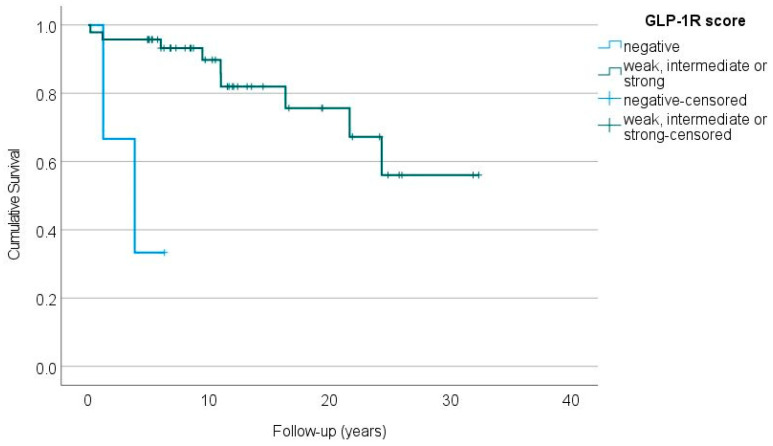
The overall survival of patients with GLP-1R-negative (blue line) vs. GLP-1R-positive (green line) sporadic insulinomas (*p* < 0.001, log-rank test).

**Figure 3 ijms-24-15164-f003:**
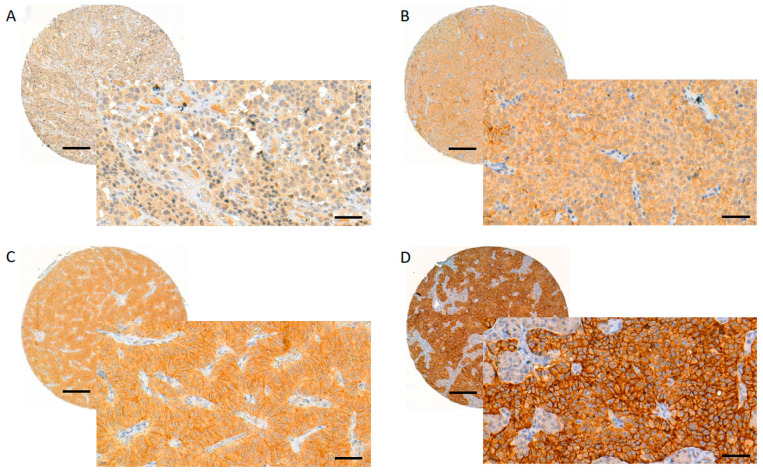
Evaluating and scoring of the GLP-1R staining. GLP-1R membranous score 0 (**A**), 1 (**B**), 2 (**C**) and 3 (**D**). Images were obtained from digitised slides with CaseViewer (3D HISTECH, Budapest, Hungary) software: whole TMA spots with magnification 8× (scale bar 200 µm), inset images with magnification 40× (scale bar 50 µm).

**Table 1 ijms-24-15164-t001:** Characteristics of 52 patients diagnosed with an insulinoma in Finland during 1980–2010, included in the analysis of GLP-1R expression.

	*n* (%)	Median (Range)
Age at surgery, years		52.7 (23.1–84.2)
Duration of follow-up after primary surgery, years		10.4 (0.2–32.4)
Sex		
Female	39 (75)	
Male	13 (25)	
Disease		
Sporadic, non-metastatic	47 (90)	
Sporadic, metastatic	3 (6)	
MEN1 ^1^-related, non-metastatic	2 (4)	
MEN1-related, metastatic	0	
Tumour diameter, mm (*n* = 46)		15 (5–60)

^1^ Multiple endocrine neoplasia type 1.

## Data Availability

The original samples and the stained slides together with the scoring data are available in the biobanks and can be accessed through the Fingenious Service at https://site.fingenious.fi/en/ (accessed on 21 August 2023).

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
