# Peer review of "Immunohistochemical Glucagon-like Peptide-1 Receptor Expression in Human Insulinomas"

_ijms, 2023, doi:10.3390/ijms242015164_

Round 1
Reviewer 1 Report (Previous Reviewer 1)
The manuscript by Vesterinen et al. “Immunohistochemical glucagon-like-peptide-1 receptor expression in insulinomas” reports a retrospective study to test GLP-1R as a biomarker for correlating the diagnosis of metastatic insulinoma and patient outcome. Overall, the goal of the study is to confirm expression of GLP-1R as a biomarker, which is repetitive using other cohorts. Major concern of the study is as the authors mentioned in the Discussion that the sample size of one arm, metastatic insulinoma is very small compared to the non-metastatic insulinoma. In addition, the analysis by IHC solely for only GLP-1R diminishes the novelty of the study. Some minor comments were added below.
The revised manuscript addressed the minor comments, however the major issues for the study still remain. More data, in addition to the IHC, would support the conclusion of the manuscript, which requires additional experiments. Current status of the manuscript doesn't look to match the aim & scope of IJMS.
1) Table 1: A unit (years) is missing to “Duration of follow-up after primary surgery.
Addressed
2) Figure legend for Fig.1 is missing.
Addressed
3) Concluding statements in each experimental result are missing, which makes harder to interpret the results.
Partially addressed
Author Response
We thank the reviewer for these comments. Since GLP-1R expression was studied earlier with the same primary antibody (Waser et al. 2015: Glucagon-like-peptide-1 receptor expression in normal and diseased human thyroid and pancreas), we wanted to confirm these results with an independent cohort. Moreover, the aim was to find biomarkers suitable for common clinical use and immunohistochemistry is a widely accessible technique.
We agree that the small number of metastatic insulinomas is the major weakness of our study, and we have now modified the conclusions, as follows: “Furthermore, the conclusive confirmation that a lack of GLP-1R expression could be used as a negative prognostic marker in insulinomas would require additional studies investigating a higher number of metastatic insulinomas.”
We do not suggest this manuscript to be published as a regular Research Article in IJMS, but as a Communication in the Special Issue “Insulinoma: From Molecular Mechanisms to Therapies”, the aims and scope of which this manuscript complies with perfectly well.
Reviewer 2 Report (New Reviewer)
Nicely written study on a well-documented series of insulinoma. Although this is not the first study to show an association between the lack of GLP1R expression and aggressive behavior in insulinoma, the confirmation of previous results is valuable, and the study is very well performed. I would recommend publication.
I have a few considerations/suggestions.
One of the MEN1 syndrome associated insulinomas lacks GLP1R expression and also shows weak insulin expression. As MEN1-syndrome patients often have multiple tumors, it would be nice to show some evidence - if possible - that this particular tumor was the symptomatic insulinoma, and not a non-functional tumor (especially as it lacks strong insulin and GLP1R expression). Did the hypoglycemic episodes resolve after resection? Was this single tumor removed by enucleation or were multiple tumors removed in a larger resection. If other tumors were present, were these stained for insulin/GLP1R?
I would consider to present overall survival in the Kaplan Meiers only for sporadic patients, as MEN1-syndrome patients have many other factors contributing to overall survival making the survival data not generalizable. For example, MEN1 patients are regularly screened for PanNETs, so it is possible the single GLP1R negative MEN1 syndrome associated insulinoma was a true aggressive case diagnosed early (before metastasizing). For patients with sporadic aggressive insulinoma, diagnosis is often made in a later stage (larger or even giant tumors, already metastasized). Therefore, while the lack of GLP1R expression may predict malignant potential of an insulinoma, it likely does not hold the same prognostic value in MEN1-syndrome and sporadic insulinoma patients.
Author Response
We thank the reviewer for the comments. The patient with GLP1R-negative MEN1-related insulinoma was diagnosed in connection with the MEN1-investigations of the patient’s sibling, and the hypoglycemic symptoms first appeared three years after the diagnosis. The single, small tumour was localized in the head of the pancreas (visible with CT, transabdominal and endoscopic ultrasound, octreotide imaging and MRI). The patient underwent pancreatico-duodenectomy, and in addition to insulinoma, no other functional or non-functional pancreatic tumour was found. The hypoglycemic symptoms resolved after the operation and no recurrence of insulinoma was detected during the long-term follow-up.
We now present in the Kaplan-Meier figure (Figure 2) the overall survival only for patients with a sporadic insulinoma.
Reviewer 3 Report (New Reviewer)
The work “Immunohistochemical glucagon-like-peptide-1 receptor expression in insulinomas” by Tiina Vesterinen et al. is devoted to the critical problem of finding biomarkers for differentiating between non-metastatic insulinomas and tumours with a more aggressive behavior. The full research cycle of this team is striking in its scale and, of course, will be of interest to International Journal of Molecular Sciences readers.
In my opinion, the work can be accepted after correcting the minor defects:
- Line 122. The authors refer to Figure 3, but Figure 2 is located under the text. There is no reference to Figure 2.
- Line 178. A superscript index is required for 111In-labelled exendin-4 derivatives.
I would also like to ask the authors to check the final documents before sending them to the journal. It was extremely inconvenient to read the editorial corrections in the text.
Author Response
We thank the reviewer for the comments. The reference to Figure 2 has now been corrected (line 122 in the Word document) and the superscript has now been placed (line 184 in the Word document).
We have also submitted a final pdf version of the manuscript, without any editorial corrections. The corrections are visible in the Word document.
Reviewer 4 Report (New Reviewer)
The manuscript by Tiina Vesterinen et al shown the histological evaluation of GLP-1 Receptor in human insulinoma. They demonstrated that the lack of GLP-1 Receptor expression was associated with impaired overall survival, larger tumour diameter, higher Ki-67 PI and weaker insulin staining. From the results obtained in these results, the authors concluded that GLP-1R expression is a useful biomarker in metastatic disease and survival in insulinoma patients. The general purpose of this study is clear. The study appears to be of interest, whereas the experiments have some concerns. In my opinion, this manuscript is not recommended for publication in its present form.
Major points
1 Results (raw data)
A first concern was lack of the immunohistochical image of Ki-67, inslin and somatostatin receptors in the insulinoma patients. We can't judge a relation and conclusion if we have no information the raw data for negative cells decided by the authors. The authors should show raw data or evidences in these points at least as supplemental figure.
2. Paragraphs
Unfortunately, I feel that this manuscript consists of too much paragraphs. Especially, Discussion section is too much. I strongly recommend you to describe in appropriate paragraphs.
3 Title.
Because the authors should emphasize and explain that the present study is a clinical study, I hope that the revised title included in the word such as "in human insulinoma" or "in insulinoma patients".
Minor points
1 Althought Ki-67 protein is a widely used marker to ascertain tumor proliferation rate based on its cellular expression during all phases of cell cycle besides G0–1, the authors need to explain the definition of Ki-67 marker in this study.
I hope these comments will be helpful.
Author Response
We thank the reviewer for these comments.
- As mentioned in the manuscript (lines 228-229 in the Word document), we presented immunohistochemical stainings and evaluation for Ki-67, insulin, and SSTR1-5 in detail in our previous study with the same patient cohort. To avoid republishing the previously published data, we omitted these data from the present manuscript.
Briefly, SSTR stainings were scored based on both membranous and cytoplasmic staining. For membranous staining we used a score 0-4 (negative (0): no staining observed; weak (1) partial membranous positivity in <10% of the tumour cell; moderate (2) partial membranous positivity in ≥10% of the tumour cells; strong (3) circumferential membranous positivity in ≥10% of the tumour cells; intense (4) >95% of the tumour cells showed a strong, circumferential staining pattern. The intensity of cytoplasmic staining was scored as negative (0), weak (1), moderate (2) or strong (3). Tumours were considered SSTR positive if a membrane pattern with a score 2 or higher was observed, or if moderate or strong cytoplasmic SSTR staining was found in ≥5% of the tumour cells. Insulin staining was considered strong if ≥50% of the neoplastic cells showed at least moderate cytoplasmic immunoreactivity. Ki-67 PI was analysed with deep-learning based Aiforia software (Aiforia Technologies, Helsinki, Finland). Please see Peltola et al. 2023: Immunohistochemical somatostatin receptor expression in insulinomas for additional information and images of the stainings. https://doi.org/10.1111/apm.13297
- We believe that dividing the discussion into multiple paragraphs makes it easier to read, and the other reviewers supported this view. Thus, we did not make changes to the paragraphs.
- The communication title has now been changed to “Immunohistochemical glucagon-like-peptide-1 receptor expression in human insulinomas”
- Ki-67 PI is indeed an essential part of NEN diagnostics and has been a part of WHO PanNEN classification since the year 2000. Moreover, it is currently the only routinely used prognostic marker in PanNENs. We emphasized the issue by adding this sentence in the discussion (lines 154-155 in the Word document): “Ki-67 PI is a cornerstone in the histopathological classification of PanNETs since it is used to grade the tumor as G1, G2 or G3, reflecting the prognosis of the disease.”
Round 2
Reviewer 4 Report (New Reviewer)
I have no concern
Author Response
We are happy to hear that after the revisions made, the reviewer has no concerns regarding our manuscript.
This manuscript is a resubmission of an earlier submission. The following is a list of the peer review reports and author responses from that submission.
Round 1
Reviewer 1 Report
The manuscript by Vesterinen et al. “Immunohistochemical glucagon-like-peptide-1 receptor expression in insulinomas” reports a retrospective study to test GLP-1R as a biomarker for correlating the diagnosis of metastatic insulinoma and patient outcome. Overall, the goal of the study is to confirm expression of GLP-1R as a biomarker, which is repetitive using other cohorts. Major concern of the study is as the authors mentioned in the Discussion that the sample size of one arm, metastatic insulinoma is very small compared to the non-metastatic insulinoma. In addition, the analysis by IHC solely for only GLP-1R diminishes the novelty of the study. Some minor comments were added below.
1) Table 1: A unit (years) is missing to “Duration of follow-up after primary surgery.
2) Figure legend for Fig.1 is missing.
3) Concluding statements in each experimental result are missing, which makes harder to interpret the results.
Author Response
Please remove the cover letter, which was erroneously attached to this response!
Responses to Reviewer 1
Comment: “The manuscript by Vesterinen et al. “Immunohistochemical glucagon-like-peptide-1 receptor expression in insulinomas” reports a retrospective study to test GLP-1R as a biomarker for correlating the diagnosis of metastatic insulinoma and patient outcome. Overall, the goal of the study is to confirm expression of GLP-1R as a biomarker, which is repetitive using other cohorts. Major concern of the study is as the authors mentioned in the Discussion that the sample size of one arm, metastatic insulinoma is very small compared to the non-metastatic insulinoma. In addition, the analysis by IHC solely for only GLP-1R diminishes the novelty of the study. Some minor comments were added below.”
Response: We thank the reviewer for this comment and agree that a small number of metastatic insulinomas is the major weakness of our study. All available primary tumour tissue samples of insulinomas diagnosed in Finland during 1980–2010 were included in this study, but due to the rarity of metastatic insulinomas, their number was limited. However, for our future studies, we plan to expand the patient cohort to cover also metastatic insulinomas diagnosed in Finland between 2011-2020.
The novelty of the current study lies in the relatively large sample of 52 insulinomas with comprehensive clinical data and long-term follow-up data of the patients. To our knowledge, this was the first study to analyse the association between immunohistochemical GLP-1R expression and overall survival. We did not analyse only GLP-1R, but also Ki-67, insulin and SSTR1-5 expression.
Comment: 1) Table 1: A unit (years) is missing to “Duration of follow-up after primary surgery.
Response: We thank the reviewer for this relevant comment. The unit “years” has now been added to the table.
Comment: 2) Figure legend for Fig.1 is missing.
Response: We thank the reviewer for this comment. We have now added the figure legend for Fig. 1, as follows:
“Immunohistochemical membranous GLP-1R expression in 52 insulinomas. The GLP-1R expression was considered strong if the staining intensity was 3+ and ≥ 50% of the tumour cells were stained. The intermediate score was given if the staining intensity was 3+ but <50% of the tumour cells were stained or if the staining intensity was 2+. The weak score was given if the staining intensity was 1+, and negative if no membranous staining was observed.”
Comment: 3) Concluding statements in each experimental result are missing, which makes harder to interpret the results.
Response: We show the concluding statements in the first sentence of each result paragraph. Please find them listed below:
2.2. GLP-1 receptor expression: All but four tumours showed at least weak expression of GLP-1R (Figure 1).
2.3. GLP-1 receptors, tumour size and Ki-67 proliferation index (PI): The lack of GLP-1R expression was associated with a larger tumour diameter and a higher Ki-67 PI.
2.4. GLP-1 receptors and insulin expression: The lack of GLP-1 receptor expression was associated with a weaker immunohistochemical staining for insulin.
2.5. GLP-1R versus SSTR expression: There was no statistically significant difference in the SSTR1–5 expression between GLP-1R positive and GLP-1R negative insulinomas.
2.6. GLP-1 receptors and metastatic or MEN1-related insulinoma: The lack of GLP-1R expression was associated with a metastatic disease (P<0.001).
2.7. GLP-1 receptors and patient outcome: The lack of GLP-1R expression was associated with a significantly impaired overall survival (Figure 3).

Reviewer 2 Report
Vesterinen et al. assessed the GLP-1 receptor expression in a set of insulinoma cases using immunohistochemistry. Within this data set they show that lack of expression of this receptor is a feature of metastatic tumors, which are generally larger, have a higher Ki-67 expression, weaker insulin staining and poorer overall survival. GLP-1 receptor expression has been therefore suggested to be a useful (excellent should be avoided in scientific texts) in estimating the metastatic potential of the tumor and the prognosis of surgically treated patients. There has been no difference in the expression of somatostatin receptor 1-5.
The manuscript would surely profit when it would be performed on a more representative sample. The authors initially claim that insulinomas metastasize in 10% of cases, while in their sample the fraction of them is 5.7%, with a total number of 3. However, authors are aware of this limitation.
Author Response
Responses to Reviewer 2
Comment: “Vesterinen et al. assessed the GLP-1 receptor expression in a set of insulinoma cases using immunohistochemistry. Within this data set they show that lack of expression of this receptor is a feature of metastatic tumors, which are generally larger, have a higher Ki-67 expression, weaker insulin staining and poorer overall survival. GLP-1 receptor expression has been therefore suggested to be a useful (excellent should be avoided in scientific texts) in estimating the metastatic potential of the tumor and the prognosis of surgically treated patients. There has been no difference in the expression of somatostatin receptor 1-5.
The manuscript would surely profit when it would be performed on a more representative sample. The authors initially claim that insulinomas metastasize in 10% of cases, while in their sample the fraction of them is 5.7%, with a total number of 3. However, authors are aware of this limitation.“
Response: We thank the reviewer for these comments. We have now replaced the word “excellent” with the word “useful” in the context of GLP-1R expression. We also agree that the small amount of metastatic insulinomas is the major weakness of our study. Our Finnish insulinoma cohort consisting of all insulinomas diagnosed in Finland during 1980–2010 (n=79) included 9 metastatic insulinomas (11.4%). In the present study, all available primary tumour tissue samples of this Finnish insulinoma cohort were included, but unfortunately only 52 samples, including only 3 metastatic insulinomas were available for analysis. However, for our future studies, we plan to expand the patient cohort to cover also metastatic insulinomas diagnosed in Finland between 2011-2020.